# Anomaly-Aware 3D Segmentation of Knee Magnetic Resonance Images

**Boyeong Woo**[1]                                                                    B.WOO@UQCONNECT.EDU.AU
**Craig Engstrom**[2]                                                                  C.ENGSTROM@UQ.EDU.AU
**Jurgen Fripp**[1,3]                                                                 JURGEN.FRIPP@CSIRO.AU
**Stuart Crozier**[1]                                                                  STUART@ITEE.UQ.EDU.AU
**Shekhar S. Chandra**[1]                                                    SHEKHAR.CHANDRA@UQ.EDU.AU

[1] *School of Information Technology and Electrical Engineering, University of Queensland, Australia*

[2] *School of Human Movement and Nutrition Sciences, University of Queensland, Australia*

[3] *Australian eHealth Research Centre, Commonwealth Scientific and Industrial Research Organisation, Australia*

**Editors:** Under Review for MIDL 2022

## Abstract

In medical imaging, anatomical structures under examination often contain anomalies or pathologies making automated segmentation challenging in these situations. Hence, the robust segmentation of anatomical structures in the presence of anomalies represents an important step within the medical image analysis field. In this work, we show how popular U-Net-based neural networks can be used for detecting anomalies in the knee from 3D magnetic resonance (MR) images in patients with varying grades of osteoarthritis (OA). We also show that the extracted information can be utilized for downstream tasks such as parallel segmentation of anatomical structures along with associated anomalies such as bone marrow lesions (BMLs). For anomaly detection, a U-Net-based model was adopted to inpaint the region of interest in images so that the anomalous regions can be replaced with close to normal appearances. The difference between the original image and the inpainted image was then used to highlight the anomalies. The extracted information was then used to improve the segmentation of bones and cartilages; in particular, the anomaly-aware segmentation mechanism provided a significant reduction in surface distance error in the segmentation of knee MR images containing severe anomalies within the distal femur.

**Keywords:** Anomaly detection, segmentation, U-Net, knee osteoarthritis, MRI

## 1. Introduction

Deep learning, particularly convolutional neural networks (CNNs), has been extremely successful in image recognition tasks and is a rapidly evolving area of research in biomedical imaging. For example, U-Net, proposed by Ronneberger et al. (2015), has become a popular CNN model for biomedical image segmentation. Çiçek et al. (2016) extended this work to dense volumetric segmentation using 3D U-Net. Efficient automated volumetric image processing is particularly important in medical imaging as manual annotation of tomographic images from techniques such as computed tomography (CT) and magnetic resonance (MR) imaging is typically expertise- and time-intensive.

In our previous study using a context aggregation network for knee MR image segmentation, it was observed that the CNN-based segmentation of images was difficult for cases with

visible, coexisting abnormalities (Dai et al., 2021). Since anomalies apart from the primary pathoanatomy of interest are frequently an inherent feature of images in clinical settings, a method to identify and quantify these "secondary" anomalies is highly desirable. In recent years, several machine learning algorithms have been proposed for automatic anomaly detection. Unsupervised methods using generative models were shown to be promising (Baur et al., 2018; Chen et al., 2020) and are particularly useful in medical imaging for which labeled data can be extremely difficult to obtain. A recent work by Pinaya et al. (2021) showed that leading-edge techniques such as transformers can also be utilized for anomaly detection and segmentation. However, application of transformers to 3D images is currently challenging due to their very high demands on data and computational resources.

In this work, we seek to detect bone marrow anomalies in MR images of the knee using models based on U-Net which are computationally less intensive than transformers. CNNs with U-Net architecture have been applied for different tasks, such as image-to-image translation (Isola et al., 2017). More recently, Liu et al. (2020) adopted a U-Net-based model to perform image inpainting and generate synthetic brain tissue intensities for a tumor region. Zavrtanik et al. (2021) proposed an anomaly detection method whereby a U-Net-based model was used to reconstruct images from partial inpaintings which were then used to localize visual anomalies in the images. The current work is similar to this idea and perform unsupervised anomaly detection on unlabeled 3D knee MR images through inpainting and lossy reconstruction. The approach is to "erase" a region of interest (ROI) from an image, which may or may not contain anomalies, and then let the network generate synthetic tissue intensities in the region without the potential anomalies. The difference between the original image and the reconstructed image can then be used to detect the anomalies. Here, we use a version of 3D U-Net for efficient volumetric image processing.

It was expected that the output from the anomaly detector can also be utilized for improving segmentation of knee MR images from patients with OA containing bone marrow anomalies. In our preliminary experiment, a 3D U-Net based on Isensee et al. (2017) was able to achieve mean Dice similarity coefficients (DSCs) for bones and cartilages in the knee joint similar to Ambellan et al. (2019) for the *OAI ZIB* dataset (dataset described in Section 2). However, without shape regularization, the surface distance errors tended to be high because U-Net outputs often contained "holes" and "noises" (false negatives/positives) due to the localized nature of the CNN-based classification. Indeed, Ambellan et al. (2019) used a combination of U-Nets and statistical shape models (SSMs), and the authors explicitly state that SSM regularization of the CNN outputs was needed to attain "anatomically plausible" segmentations, which is consistent with our preliminary finding. In particular, it was difficult to achieve a good segmentation with a conventional U-Net when there was a large anomaly in the image (see Figure 5). Therefore, we also propose an anomaly-aware segmentation mechanism for U-Net which is more robust in the presence of anomalies.

Overall, this work will show (1) using 3D U-Net-based CNNs for anomaly detection and (2) improving segmentation of knee MR images using anomaly-aware mechanism.

## 2. Materials and Methods

This study used a publicly available knee MR image dataset *OAI ZIB*, used previously by researchers at Zuse Institute Berlin (ZIB) (Ambellan et al., 2019). The dataset consists of

507 MR imaging examinations from the Osteoarthritis Initiative (OAI) database ([https://oai.nih.gov](https://oai.nih.gov)) for which reference segmentations were carried out by experienced users at ZIB starting from a model-based auto-segmentation. The images were acquired from the right knee at baseline using a 3D weDESS (water excitation double-echo steady state) sequence with $160\times384\times384$ voxels. Segmentation labels consist of the femoral and tibial bones and cartilages. The dataset covers the full spectrum of OA grades. Other details of the dataset can be found in Ambellan et al. (2019). Ethics approval for collection of data was obtained by the OAI and the participating clinical sites (Nevitt et al., 2006).

Figure 1 shows the overall pipeline for our method. In Part 1, we erased regions of femur and tibia in the images and then inpainted these regions using a U-Net-based model. In Part 2, we used the outputs from Part 1 and another U-Net-based model to replace anomalous regions in the original images with close to normal appearances. In Part 3, the information extracted from Part 2 was used to improve the segmentation of bones and cartilages. Each part will be explained in the following subsections. The dataset was randomly split into 5-fold cross-validation sets and maintained for all components of Parts 2 and 3 for evaluation of segmentation performance. See Appendix A for further details on implementation.

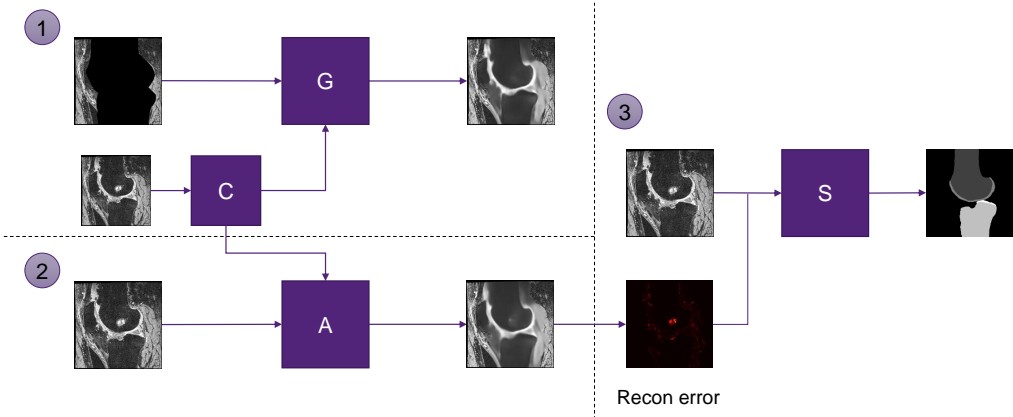

Figure 1: Overall pipeline including anomaly detection and the downstream task. Parts **1** and **2** are the anomaly detection models shown in Figure 2. Part **3** is the downstream task using the model shown in Figure 3.

## 2.1. Anomaly detection using masked images

Using the reference segmentation masks available in the *OAI ZIB* MR dataset, the profiles of the femur and tibia and their surrounding areas were erased from the MR images. The masks were dilated by 50 pixels in all directions before being applied to the images to include the surrounding areas. These masked images were then used as input to a neural network with a 3D U-Net-like architecture, which was trained to recover the original image, i.e. to inpaint the erased area. This network will be referred to as $G$.

Since the erased profile was relatively large, an image compressor $C$ was also added to assist with the inpainting. This small network was trained simultaneously with $G$ to compress the original MR image, and the compressed image was fed into the decoder part of $G$.

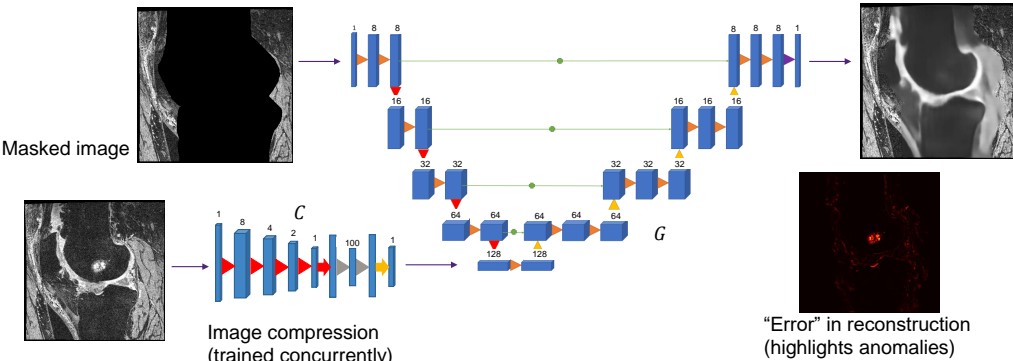

($a$) Network $G$ regenerating the original images from masked images through inpainting and decoding of compressed images.

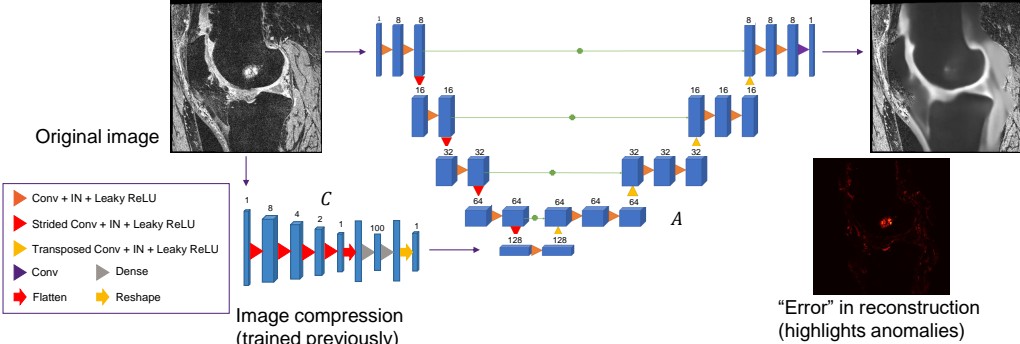

($b$) Network $A$ trained using the images generated by $G$ as the target output given the original images.

Figure 2: The two anomaly detection networks with a 3D U-Net-like architecture. *IN: Instance Normalization.*

The overall model structure is shown in Figure 2($a$). The loss function for training $G$ and $C$ was the mean squared error (MSE) between the original image $x$ and the regenerated image $G(x)$: $\mathcal{L}_G = ||x - G(x)||_2^2$. The model was expected to recover most of the image but not the anomalous regions, and therefore, the difference between the original and reconstructed images could be used to highlight the anomalies, i.e. the region with larger difference is more likely to be anomalous.

## 2.2. Anomaly detection using the original images

A limitation of the above model $G$ is that it requires a segmentation mask to generate input for the network. Therefore, another network $A$ was trained to take the original images (without masking) as the input and mimic the outputs of $G$. The network architecture of $A$ was the same as $G$, and the image compressor $C$, trained previously with $G$, was added here as well. The overall model structure is shown in Figure 2($b$).

The loss function for training $A$ was the MSE between the output from the previous network $G(x)$ and the output from the current network $A(x)$, plus the MSE between the

respective error images $\mathrm{E}(G(x)) = (x - G(x))^2$ and $\mathrm{E}(A(x)) = (x - A(x))^2$ to further guide the model: $\mathcal{L}_A = ||G(x) - A(x)||_2^2 + ||\mathrm{E}(G(x)) - \mathrm{E}(A(x))||_2^2$. It was expected that this model would produce outputs that are very similar to the previous model but would have the advantage of not requiring segmentation masks. It would detect anomalies when given the original images only.

### 2.3. Downstream task: Anomaly-aware segmentation

The error images from $A$, which highlight the anomalies in the original MR images, were utilized to construct a segmentation model which could handle anomalies. The segmentation network $S$ was again a 3D U-Net, but with deep supervision as in Isensee et al. (2017), which helps to accelerate training. A commonly used loss function for medical image segmentation is a multiclass Dice loss, as defined below:

$$\mathcal{L}_{DSC} = 1.0 - \frac{2}{|K|} \sum_{k \in K} \frac{\sum_i u_{i,k} v_{i,k}}{\sum_i u_{i,k} \sum_i v_{i,k}}. \tag{1}$$

Here, $u$ is the softmax output of the network and $v$ is the one-hot encoded ground truth segmentation map. $K$ is the number of classes. $u_{i,k}$ and $v_{i,k}$ denote the softmax output and ground truth label, respectively, for class $k$ at voxel $i$.

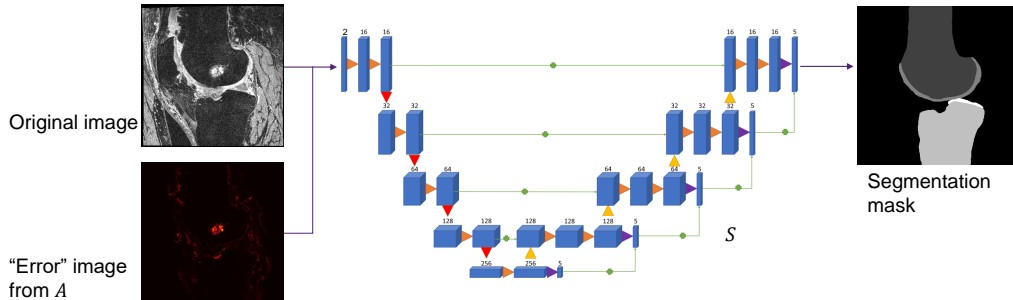

Figure 3: The anomaly-aware segmentation network $S$ based on 3D U-Net with deep supervision.

Inspired by Nie and Shen (2020), which used a difficulty-aware attention mechanism, we added a focal cross-entropy loss where the focal weights were given by the error images:

$$\mathcal{L}_{FCE} = - \sum_i \sum_{k \in K} F_i v_{i,k} \log u_{i,k} \tag{2}$$

where $F = 1.0 + \beta \mathrm{E}(A(x))$. The total loss for the segmentation network was then: $\mathcal{L}_S = \mathcal{L}_{DSC} + \alpha \mathcal{L}_{FCE}$. Here, $\alpha = 10.0$ and $\beta = 99.0$ were used. Using this loss, the network was trained to pay more attention to the voxels that were found to be anomalous and hence likely to be difficult for the segmentation network to classify. The error images were also used as an additional input (Figure 3). Since it is assumed that segmentation masks are not available for test images, only the outputs from $A$, which did not require segmentation masks, were used for both training and testing.

## 3. Results

Figure 4 shows two example outputs from the anomaly detection networks. The input MR images of the knee have some visible anomalies of the distal femur such as BMLs and osteophytes. It can be seen that the network outputs are lossy reconstructions of the input images with the cancellous bone anomalies mostly removed from the images. The outputs from $A$ are a bit closer to the original images compared to the outputs from $G$ but still have most of the anomalies blurred out. The last column of Figure 4 shows the reconstruction errors highlighting the anomalous regions within the cancellous bone of the distal femur.

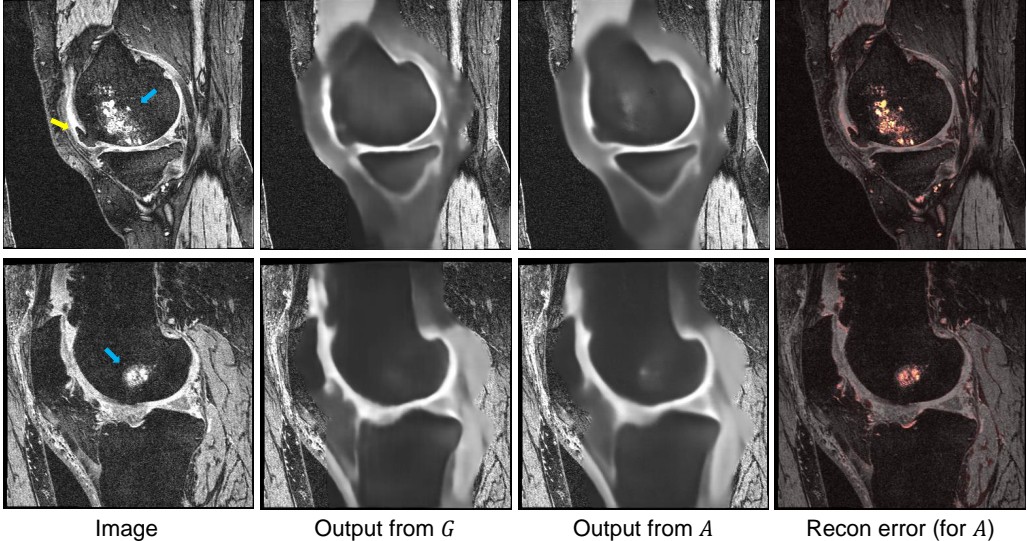

| Image | Output from $G$ | Output from $A$ | Recon error (for $A$) |

Figure 4: Example outputs from $G$ and $A$. The last column shows the error images (color-mapped and overlaid on the input images) highlighting the difference between the input image and the output from $A$. Blue arrow: BMLs; Yellow arrow: osteophyte.

Figure 5 shows two example outputs from the anomaly-aware segmentation network $S$ and compares them with the outputs from U-Net without the anomaly-aware mechanism. As noted in Section 1, the conventional U-Net (similar to Isensee et al. (2017), using multiclass Dice loss only) performed well in terms of DSCs for most of the *OAI ZIB* images (See Figure A1 for example outputs where it performed well), but failed on some cases with severe abnormalities. The anomaly-aware method was found to be more robust against these difficult cases, resulting in a noticeable improvement in the quality of the segmentation of bone volume for the femur and tibia (Figure 5).

Table A1 shows some quantitative results for the segmentation task. There was not a substantial improvement in the mean DSCs, but some statistically significant improvements in surface distances. In particular, the average surface distance (ASD) of femoral bone was significantly improved (p-value $\ll 0.001$ using t-test), and the Hausdorff distances (HDs) were reduced by 46–86% for all of the classes (See Appendix B for the definitions of DSC, ASD, and HD). See also the boxplot in Figure 6 to compare the distributions of HDs.

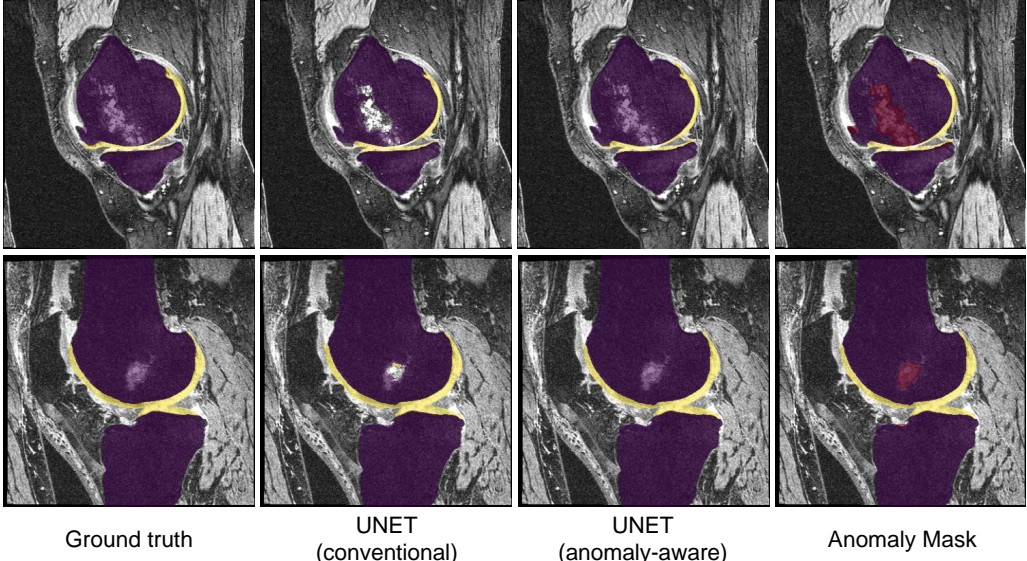

Figure 5: Example outputs from $S$. The masks are overlaid on the input images (purple: femoral and tibial bones; yellow: femoral and tibial cartilages). Notice the segmentation errors from conventional U-Net, which were largely corrected with the anomaly-aware mechanism. The anomaly mask in the last column (red) was produced by combining the outputs from $A$ and $S$ to segment the abnormalities within the cancellous bone. See Figure A2 for a more detailed view.

## 4. Discussion and Conclusion

The last column of Figure 4 shows that the anomaly detection networks appear capable of highlighting visible anomalies from MR images of the knee. These autoencoder-based networks also reconstruct the images with most of the BMLs removed and therefore can be used to show what the femoral and tibial bones likely looked like if they had no BMLs. The main limitation for this is that the quality of the reconstructed images are rather poor. This is complex because it is actually easy for convolutional autoencoders to reconstruct images with minimal reconstruction error, but for anomaly detection, the reconstruction needs to be lossy since the anomalies should be removed, which means we might need to compromise on image quality. Nevertheless, a useful future work would include improving the model to make the images look more anatomically realistic. A possible approach would be to modify the image compressor, for example by using a more sophisticated model such as a conditional encoder or vector quantization (van den Oord et al., 2017).

The result of the downstream task demonstrated that the anomaly detectors are also useful for improving segmentation of overall bone volume from MR images of the knee in OA patients with visible anomalies. The conventional U-Net was already capable of achieving high DSCs, but it was difficult to achieve good HDs without any shape regularization or post-processing. The anomaly-aware attention mechanism provided a substantial improvement in HDs in addition to a visible improvement in the quality of segmentation. It

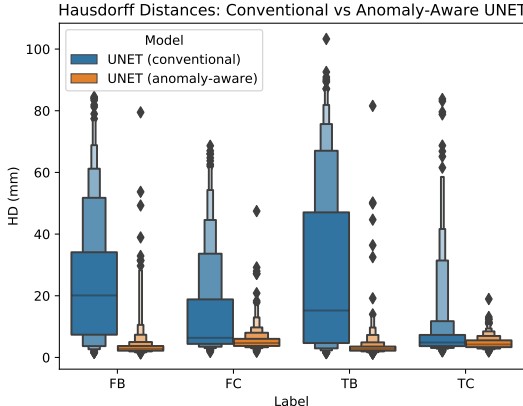

Figure 6: Boxplot comparing Hausdorff distances (HDs) for the conventional U-Net (similar to Isensee et al. (2017)) and the anomaly-aware U-Net (proposed) evaluated using 5-fold cross-validation. HDs for the anomaly-aware method are much smaller overall. *FB: femoral bone; FC: femoral cartilage; TB: tibial bone; TC: tibial cartilage.*

appeared that the new method also performed some shape correction probably because the additional information provided to the network guided the network to focus its attention to the ROI, and the model was less sensitive to anomalies and noise in the images. While this study compared the proposed method with the conventional U-Net only, follow-up work with expanded experiments and analyses, including ablation studies, will provide evaluation on the effectiveness of each component and comparison with state-of-the-art methods.

The overall segmentation pipeline ($A$ and $S$) provides two main pieces of information: anatomical structure and anomalies in the image. By combining the two outputs from $A$ and $S$, we can also segment co-existing pathologies in the MR images such as BMLs and osteophytes in an unsupervised manner (See Figure A2). A main limitation in the current study is a lack of quantitative analysis for the anomaly localization, primarily due to a lack of ground truth annotation of the anomalies. Future work comparing the automatically detected anomalous regions with expert manual segmentations will allow dedicated quantitative analysis. Also, while the cross-validation split was maintained for $A$ and $S$, the split was not applied for $G$ in the current study which may limit generalizability; therefore, a follow-up study testing these models on images outside the *OAI ZIB* dataset would be beneficial for evaluating generalizability. In Figure 1, it can be noted that the models $A$ and $S$ actually form a linear pipeline. Therefore, future work could investigate combining $A$ and $S$ into a single multi-task model to perform anomaly detection and segmentation simultaneously. In addition, since we can detect and segment pathologies, a future work may also include another downstream task such as classification of OA grades.

In summary, this work demonstrated how a simple U-Net-like neural network can be used for detecting bone anomalies in knee MR images. Moreover, it showed how the detected anomalies can be further utilized for downstream tasks such as segmentation. Future works are expected to show additional improvements and applications of the anomaly detection and anomaly-aware segmentation models in medical imaging.

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

## Appendix A. Implementation Details

The three networks $G$, $A$, and $S$ (Figures 2 and 3) are all based on 3D U-Net (Çiçek et al., 2016). Overall, the networks consist of 5 levels of resolution with 4 progressive downsamplings using strided convolutions. There are two convolution blocks at each level in the contracting path, where each convolution block consists of a 3D convolution layer with a kernel size of $3 \times 3 \times 3$ followed by an instance normalization (Ulyanov et al., 2016) and leaky rectified linear unit (ReLU) activation function with a negative slope coefficient of 0.1. The number of feature maps are progressively increased as the resolution is decreased.

In the expansive path, the resolution is progressively recovered using transposed convolutions with a stride of 2. The feature maps from the encoder at each resolution are transferred using element-wise summation. The combined feature maps then pass through two more convolution blocks before another transposed convolution for upsampling. When the original image resolution is recovered, the final output convolution layer with a kernel size of $1 \times 1 \times 1$ and softmax activation is applied.

Several previous works on volumetric segmentation using 3D CNNs (typically U-Net-like) have employed a technique referred to as "deep supervision" (Kayalibay et al., 2017; Isensee et al., 2017; Raj et al., 2018). The main idea of deep supervision is to provide integrated direct supervision to the hidden layers, rather than providing supervision only at the output layer (Lee et al., 2015). Here, deep supervision was applied to both the conventional U-Net and the anomaly-aware U-Net for the downstream task (Section 2.3). Deeply supervised U-Net produces secondary segmentation maps at deeper levels of the network and combines them with the final segmentation map via upsampling and element-wise summation (Figure 3). Deep supervision speeds up convergence by encouraging deeper layers to produce improved segmentation results (Kayalibay et al., 2017).

The neural networks were implemented using Tensorflow (Abadi et al., 2015) version 2.4 with Keras API (http://tensorflow.org/guide/keras). For the downstream task (Section 2.3), the "mixed precision" policy in Keras API was used for both the conventional U-Net and the anomaly-aware U-Net to overcome memory limitation when training 3D CNNs with a large number of feature maps. Mixed precision refers the use of 16-bit floating-point type in parts of the model during training to make it use less memory. Usually, the output layer (softmax layer) is kept in the 32-bit type for numeric stability. Despite using lower precision data type, numerical instability was not observed in these segmentation models.

The images were Z-normalized, clipped at [-5, 5], and subsequently rescaled to [0, 1] for network input. The models were trained using the Adam optimizer (Kingma and Ba, 2014) with a learning rate of 5e-4, batch size of 1, and for 50 epochs. They were trained on a high-performance computer with NVIDIA Tesla V100-SXM2-32GB.

## Appendix B.  Evaluation Metrics

Segmentation performance was evaluated using Dice similarity coefficient (DSC), average surface distance (ASD), and Hausdorff distance (HD; also known as maximum surface distance):

$$DSC = \frac{2|B \cap A|}{|B| + |A|}, \tag{3}$$

$$ASD = \frac{1}{|\partial(A)| + |\partial(B)|} \left( \sum_{a \in \partial(A)} \min_{b \in \partial(B)} ||a - b||_2 + \sum_{b \in \partial(B)} \min_{a \in \partial(A)} ||b - a||_2 \right), \tag{4}$$

$$HD = \max \left( \max_{a \in \partial(A)} \min_{b \in \partial(B)} ||a - b||_2, \max_{b \in \partial(B)} \min_{a \in \partial(A)} ||b - a||_2 \right). \tag{5}$$

Here, $A$ and $B$ denote the set of positive voxels in the ground truth segmentation map and the predicted segmentation map, respectively, and $\partial(\cdot)$ denotes the boundary of the segmentation set.

## Appendix C.  Table

Table A1: The segmentation performance in terms of mean DSC, ASD, and HD for conventional U-Net (similar to Isensee et al. (2017)) and the anomaly-aware method (proposed) evaluated using 5-fold cross-validation. The superscript ** denotes a significant difference at p-value ≪ 0.001 with t-test. There was a borderline significance for ASD of TB at p-value = 0.05, denoted by *. Other p-values were > 0.1. *FB: femoral bone; FC: femoral cartilage; TB: tibial bone; TC: tibial cartilage.*

| Model | Class | DSC (%) | ASD (mm) | HD (mm) |
|-------|-------|---------|----------|---------|
| **UNET** (conventional) | **FB** | 98.7±0.33 | 0.25±0.09 | 24.18±19.96 |
| | **FC** | 89.7±2.91 | 0.26±0.08 | 14.06±15.04 |
| | **TB** | 98.7±0.34 | 0.25±0.37 | 27.63±26.23 |
| | **TC** | 85.8±4.16 | 0.27±0.10 | 8.75±12.13 |
| **UNET** (anomaly-aware) | **FB** | 98.7±0.30 | **0.22±0.06**\*\* | **4.05±5.94**\*\* |
| | **FC** | 89.5±2.67 | 0.26±0.07 | **5.58±3.69**\*\* |
| | **TB** | 98.7±0.32 | 0.22±0.14\* | **3.82±5.63**\*\* |
| | **TC** | 86.0±4.00 | 0.26±0.09 | **4.74±2.03**\*\* |

## Appendix D.  Extra Discussion

It is also worth noting that there were errors in the "ground truth" segmentations of specific anatomical structures (e.g. inaccurate identification of bone boundaries); in several cases, the proposed method appeared to have produced a more authentic segmentation of the osteophytes. See Figure A3 for an example where the ground truth segmentation appeared less accurate around an osteophyte. This suggests that the CNN-based method may potentially perform better than the traditional methods if trained well.

**Appendix E. Extra Figures**

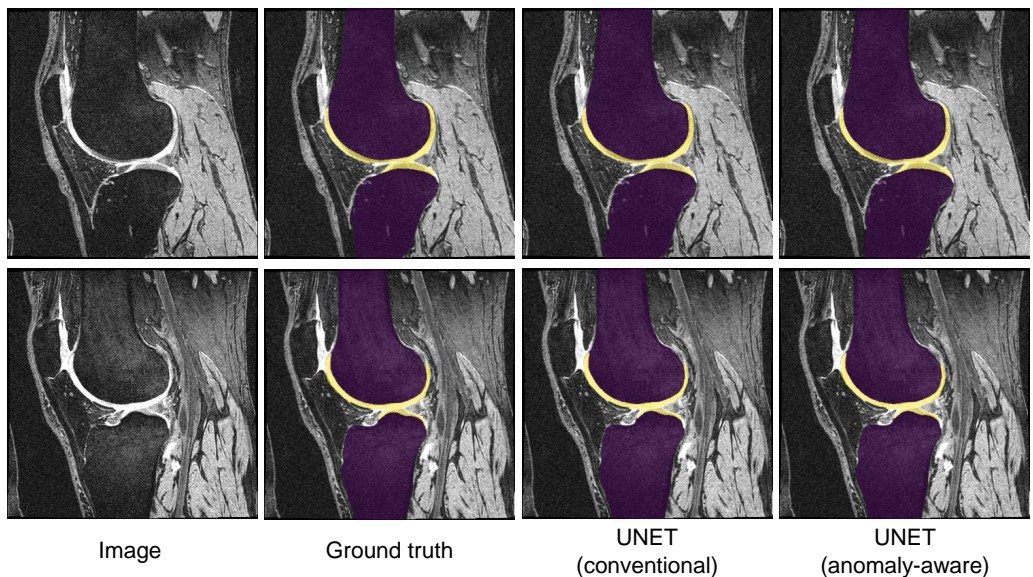

|  |  |  |  |
|---|---|---|---|
| Image | Ground truth | UNET (conventional) | UNET (anomaly-aware) |

Figure A1: Example segmentation outputs for images with little to no visible anomalies. The masks are overlaid on the input images (purple: femoral and tibial bones; yellow: femoral and tibial cartilages). Both the conventional UNET and the anomaly-aware method (proposed) produced good segmentation masks.

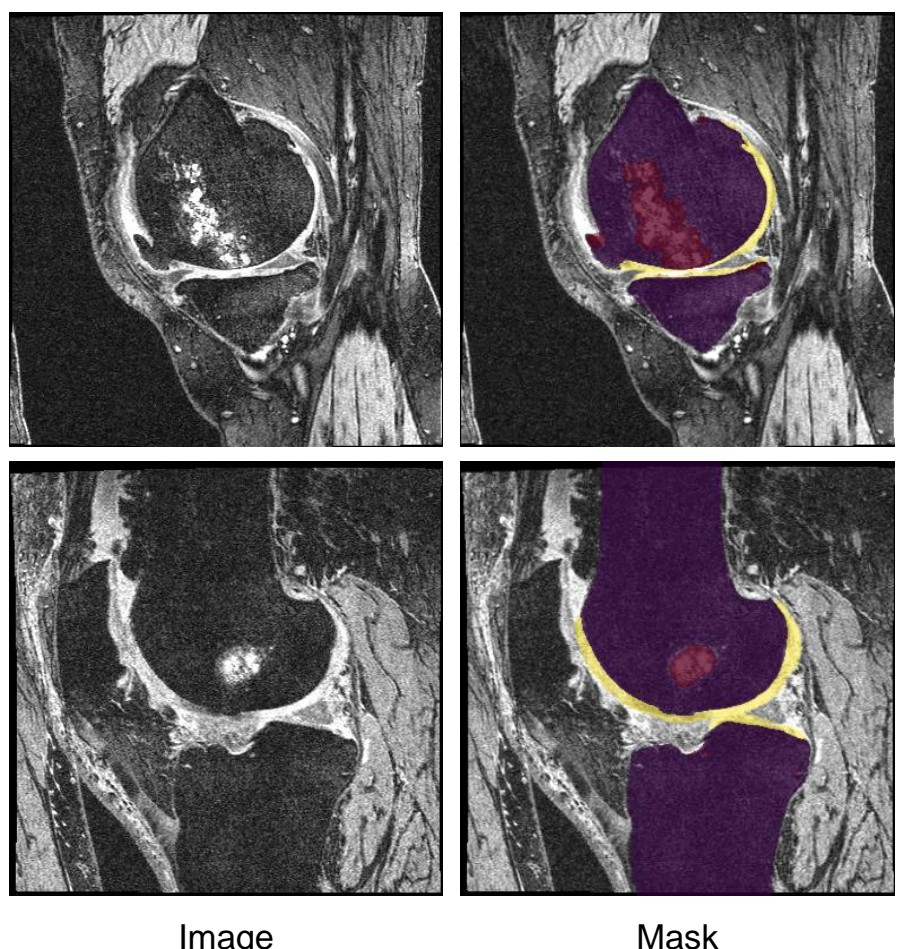

Image           Mask

Figure A2: Combining the outputs from $A$ and $S$ to segment the abnormalities within the bone tissue such as bone marrow lesions and osteophytic regions (red mask). The error image from $A$ was smoothed using a Gaussian filter to remove noise and then thresholded to produce the mask.

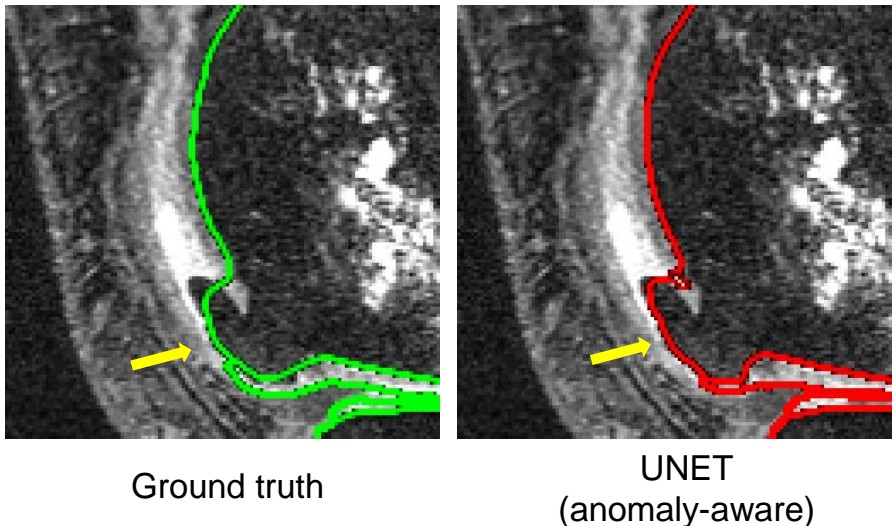

**Ground truth**

**UNET
(anomaly-aware)**

Figure A3: An example image (zoomed-in view) showing an osteophyte, indicated by the yellow arrow. The ground truth segmentation (left) misses the detail around the boundary of the osteophyte. The automated method (right) seems to have produced a more accurate segmentation in this case.

