# OpenReview forum: "Anomaly-Aware 3D Segmentation of Knee Magnetic Resonance Images"
_MIDL.io/2022/Conference — MIDL 2022_

### Official Review · Reviewer_YpcE · 2022-01-17

**Confidence:** 4
**Preliminary Rating:** 4
**Recommendation:** Poster

**Summary:**

The authors proposed a novel anomaly-aware network for knee MRI segmentation. The network is trained to focus on anomalous region by utilizing the reconstruction error. The proposed method generates segmentation results with better ASD/HD performance compared with conventional U-Net on a large-scale public dataset.

**Strengths:**

- Generally, the paper is well written, and the proposed method is novel and interesting.
- Evaluation on a large-scale public dataset with better ASD/HD performance compared with conventional U-Net.

**Weaknesses:**

- In the abstract, the authors state several challenges for 3D medical image segmentation. However, some challenges like lack of large image datasets, is not the scope of this work. The abstract should be a short summary of novelty and contribution of proposed methods. However, the abstract of current version seems tedious and should be reorganized.
- “All of these U-Net-based models were fully volumetric convolutional neural networks, allowing for efficient 3D image processing” Here what does efficient 3D image processing mean?
- Fig.2 (b) is very helpful for understanding. I think it would be better to introduce of the pipeline in Fig.2 (b) at the beginning of this section to give an overall impression, before introducing the details of each subsets (A,G,C,S).
- The experiments are relatively simple. The authors only compare their method with conventional U-Net, instead of other state-of-the-art methods. Lack of ablation experiments to evaluate the effectiveness of each subsets.
- Some typos: Section1. shape regularisation -> regularization


**Deanonymize Review:**

no

**Paper Type:**

methodological development

**Questions To Address In The Rebuttal:**

I would like the authors to address the points of the weaknesses above.


- In the abstract, the authors state several challenges for 3D medical image segmentation. However, some challenges like lack of large image datasets, is not the scope of this work. The abstract should be a short summary of novelty and contribution of proposed methods. However, the abstract of current version seems tedious and should be reorganized.
- “All of these U-Net-based models were fully volumetric convolutional neural networks, allowing for efficient 3D image processing” Here what does efficient 3D image processing mean?
- Fig.2 (b) is very helpful for understanding. I think it would be better to introduce of the pipeline in Fig.2 (b) at the beginning of this section to give an overall impression, before introducing the details of each subsets (A,G,C,S).
- The experiments are relatively simple. The authors only compare their method with conventional U-Net, instead of other state-of-the-art methods. Lack of ablation experiments to evaluate the effectiveness of each subsets.

**Special Issue:**

no

---

### Official Review · Reviewer_uXUe · 2022-01-23

**Confidence:** 4
**Preliminary Rating:** 3
**Recommendation:** Poster

**Summary:**

The authors describe a method for detecting anomalies in MR images of the knee. Their method uses a 3D U-Net to inpaint removed regions in the original MR image. The anomalies are then determined by the reconstruction errors between inpainted and original image. The detected anomalies are used as additional information for segmentation into bone, cartilage, and background. The authors show that the Dice score does not change compared to a baseline, but the Hausdorff distance decreases significantly.

**Strengths:**

The paper is clearly structured and well written.
The idea of using detected anomalies to improve the downstream segmentation task is clearly valuable.
The significant reduction in the Hausdorff distance shows that the method provides better segmentation.

**Weaknesses:**

No description of the dataset is given. The dataset is public and referenced, but there is no indication of the split between training, testing and validation data.

Is the split maintained over all models? Is a training sample for model G always a training sample for model A and S?
If the split for is not the same for G and A could information leak from G's training samples into A's validation/test samples?
Is there any kind of data augmentation? If not, why?

---

The authors state that the "The anomaly detection networks achieved their main objective to ipso facto highlight cancellous bone anomalies [...]" with no further proof then some example images.
While the sample images look good, there is not quantitative measurement of the detection quality.
Without such a measurement, it is hard to claim with certainty that the task has been achieved.

---

Neither inpainting using a U-Net nor using the reconstruction error to detect anomalies is a new method. See the referenced paper (Liu et al., 2020) or [1] for another example. Therefore I can not see methodological development on this steps.

Using these anomalies in the downstream task to improve segmentation results seems to be the new and more valuable contribution.
I think the focus should be on improving segmentation and less on anomaly detection.

[1] Zavrtanik, V., Kristan, M., & Skočaj, D. (2021). Reconstruction by inpainting for visual anomaly detection. Pattern Recognition, 112, 107706. https://doi.org/10.1016/j.patcog.2020.107706

**Deanonymize Review:**

yes

**Detailed Comments:**

- If the data is already public it would be good to see the source code.
- If the improvement in the Hausdorff distance compared to the baseline in the segmentation task comes from the detected anomalies, the baseline should be as good as the proposed method on images with no anomalies.
The input of detected anomalies should be empty (since there are none), and therefore both methods should get the same information.
Have you compared the results for images containing no anomalies for both methods?


**Final Rating After The Rebuttal:**

4: Weak Accept

**Justification Of The Final Rating:**

The authors have addressed most of my concerns, justified their decisions, and significantly improved their paper.
While I still have some minor concerns (like information leakage from model G to A and S), I think the paper can be accepted.

**Paper Type:**

validation/application paper

**Questions To Address In The Rebuttal:**

- The claim "The anomaly detection networks achieved their main objective to ipso facto highlight cancellous bone anomalies [...]" should be supported with facts or removed.
- At least a minimal description of how training, testing, and validation data are handled should be provided.

**Special Issue:**

no

---

### Meta-Review · Area_Chair_T6ZM · 2022-02-18

**Recommendation:** Accept (Poster)
**Confidence:** 4

**Metareview:**

The paper has received 2 reviews from which one reviewer has only provided a final evaluation after rebuttal. Therefore, there might be an uncertainty in the rating left. However, I would recommend acceptance of the paper as a poster, since the authors have justified their decisions, and significantly improved their paper. Results and discussion seem to match with the proposed novel method.

---

### Decision · Program_Chairs · 2022-02-28

Accept